# EU space security–An 8-Step online discourse analysis to decode hybrid threats

João Reis[1,2,3,4]*

1 Industrial Engineering and Management, Faculty of Engineering, Lusófona University, Lisbon, Portugal, 2 Department of Military Sciences, Military Academy, CINAMIL, Lisbon, Portugal, 3 RCM2+ Research Centre for Asset Management and Systems Engineering, Lusófona University, Lisbon, Portugal, 4 Research Unit on Governance, Competitiveness and Public Policies (GOVCOPP), Campus Universitário de Santiago, Aveiro, Portugal

* joao.reis@ulusofona.pt, joao.reis.academic@gmail.com

**Data Availability Statement:** Articles included in this review are fully accessible through databases - Web of Science and Elsevier Scopus.

**Funding:** The author(s) received no specific funding for this work.

## Abstract

Space security has emerged as a concern for the European Union (EU), given that space systems have become integral to ensuring the safety of all European society. This strategy reflects the interaction of geopolitical dynamics and the rising specter of hybrid threats. However, grappling with hybrid threats targeting the EU space presents distinct challenges, primarily owing to their elusive nature. Hence, our objective is to develop practical methodologies to identify and mitigate such threats effectively. To meet this objective, we propose an innovative 8-step process. This approach streamlines the systematic identification and analysis of discourses of hybrid threats within online platforms dedicated to EU space discussions. Our methodology underwent rigorous scrutiny, including a comprehensive literature review that prioritized peer-reviewed manuscripts from sources such as Web of Science and Elsevier Scopus. Additionally, we selected documents from Google Scholar to ensure comprehensive coverage of diverse scholarly contributions, enriching the depth of our analysis. Our research yielded a conceptual framework for Online Discourse Analysis (ODA) tailored to evaluate hybrid threats targeting EU space defense and security. The results highlight the importance of leveraging advanced ODA techniques to deepen our understanding of emerging threats. In conclusion, we advocate for the adoption of these innovative methods to enhance the robustness of EU space defense strategies in the face of evolving security scenarios. The establishment of the 8-step ODA framework marks a pivotal milestone, offering a structured approach to deciphering hybrid threats. Looking ahead, we aim to empirically validate this framework by creating and deploying custom-tailored software designed to identify potential hybrid threats jeopardizing the security of European space assets. Through continued research and practical implementation, we endeavor to fortify the EU's defenses against emerging threats in the space domain.

## 1. Introduction

Over the past six decades, the landscape of the space economy has undergone a remarkable evolution. It has shifted from being the exclusive purview of a handful of spacefaring nations

**Competing interests:** The authors have declared that no competing interests exist.

to encompass the active participation of over 60 countries worldwide, spanning a diverse spectrum of space-related activities [1]. This growth trajectory not only endures but also exhibits clear indications of gaining momentum. Thus, in an era where space technology's significance continues to burgeon, the protection of Europe's invaluable space assets has rightfully assumed a position of utmost significance.

As nations earnestly pursue their interests within space, the emergence of hybrid threats adds a multifaceted and dynamically shifting dimension to the challenge. In this context, the vulnerabilities within space operations possess the potential to compromise vital elements such as satellites, ground stations, and communication links, thereby triggering a cascade of disruptive effects [2]. Consequently, our article undertakes an investigation into the security of the European Union's (EU) space domain. By harnessing advanced Online Discourse Analysis (ODA) techniques, our inquiry seeks to uncover and comprehend the nuanced facets of hybrid threats, shedding light on the dynamics they entail.

Acknowledging the subject´s importance, the utilization of ODA techniques to exploit hybrid threats remains a relatively unexplored domain within the scientific inquiry. Yet, a noteworthy breakthrough was achieved by Dragos et al. [3], who conducted a comprehensive analysis. This analysis encompassed supervised, unsupervised, and semantic approaches, all applied to the examination of social data flows. Their study sheds light on various techniques employed to unveil latent patterns within socially generated data collections. These collections are characterized by an abundance of unstructured text, a fusion of modalities, and the potential for inaccurate spatiotemporal markers.

The work of Dragos et al. [3] also investigates a case study presented by Bieteniece et al. [4] regarding online advertising detection. Bieteniece et al. [4] aimed to showcase instances where one of the objectives was to identify Russian influence. This highlights the prevalence of online propaganda, disinformation, and biased narratives aimed at manipulating campaigns, social networks, and platforms. Tactics such as the use of virtual bots to sway public opinion, amplify or suppress political content, and spread disinformation, hate speech, violence, and fake news are commonplace. A well-known example is the 2016 US presidential elections, which involved attempts to polarize public opinions.

However, the specific case under scrutiny involves the application of filters to a Twitter dataset, with the goal of detecting foreign influence campaigns in Canada, in October 2019. Two simple filters were developed to spot suspicious tweets of Russian origin. One filter examined the content, while the other focused on users. The first filter consisted of a list of 200 websites sourced from www.propornot.com, which identifies sites propagating Russian propaganda. The second filter identified authors associated with the Internet Research Agency (IRA), a group labeled as Russian Trolls by Twitter. This dataset was made public by Twitter in 2018. In their study, Bieteniece et al. [4] unveiled a substantial BOTNET (i.e., a collection of computers/devices that have fallen victim to malware and are controlled by an attacker) comprising numerous BOTs (i.e., infected devices) engaged in various matters. Remarkably, they identified at least two BOTs that actively contributed to all the wedge issues. Interestingly, the application of the IRA filter yielded no outcomes. Bieteniece et al. [4] speculates that the compromised status of the Russian Troll handles, following their disclosure by Twitter, has rendered them inactive. An alternative solution to the user-based filter could involve the utilization of BOT handles. This alternative approach demonstrated remarkable efficacy, yielding positive outcomes. Due to the scholarly nature of this article, it is crucial to provide divergent perspectives. In this context, the work of Menshikov & Neymatova [5] asserts that within the framework of escalating anti-Russian information warfare and the intensifying ideological conflict, active information support for Russia's foreign policy assumes heightened significance. The authors endeavor to validate their stance by showcasing historical instances of the

US involvement in information warfare against Russia. Additionally, they contend that the US has progressively adopted more sophisticated tactics over time, enlisting the EU and NATO as collaborative partners in these efforts.

As closing remarks, and based on our interpretation, it becomes crucial to collectively acknowledge the necessity of addressing hybrid threats on a global scale, while upholding the autonomy and territorial integrity of each nation. This approach should extend to encompass challenges linked to space-related matters. The endeavor to tackle these concerns and establish proficient countermeasures against hybrid threats, all the while nurturing the security of individual countries, emerges as a shared mutual concern that should command the attention of all involved parties.

Returning to the research of Dragos et al. [3], their findings underscore that relying solely on Machine Learning (ML) models and semantic approaches within social media data mining exposes vulnerabilities. The transformative impact of social media on global connectivity has reshaped communication, idea exchange, and virtual community organization. Gaining insights into online behaviors and processing digital content carries significant implications for security applications. Nevertheless, the sheer volume of data, its inherent noise, and the rapid shifts in topics present substantial challenges that hinder the effectiveness of classification models and the applicability of semantic frameworks.

The existing gap in the literature revolves around the limited exploration of ODA within the specific context of identifying and preempting actions driven by hybrid threats. While ODA has exhibited effectiveness in diverse contexts, it is notable an absence of literature discussing its application in both safeguarding space assets and detecting hybrid threats. With the aim of bridging this void, we have formulated two research questions (RQ), which we outline below:

RQ1. How can Online Discourse Analysis (ODA) techniques be effectively integrated into the space domain to identify, analyze, and mitigate hybrid threats, taking into consideration the evolving landscape of space-related activities and the diverse range of potential threats?

RQ2. What are the limitations and challenges associated with current hybrid threat detection methods in the context of space operations, and how can advanced ODA techniques be harnessed to overcome these limitations and provide a more comprehensive understanding of emerging threats?

In addressing RQ1, we will elucidate the utilization of ODA techniques within the EU space. This will involve outlining an elaborated 8-step process aimed at proficiently recognizing, scrutinizing, and mitigating hybrid threats. Transitioning to RQ2, our aim is to introduce a comprehensive conceptual framework. In addition, we will engage in a comprehensive discourse concerning the use of the ODA technique, investigating the various challenges they pose as well as the opportunities they present.

This article adheres to the IMRaD or IMRyD format (Introduction, Methods, Results and Discussion), a widely recognized structure for organizing scientific articles. Additionally, we have included two supplementary sections: one providing a theoretical framework and the other offering a conclusive summary focusing on the originality and contributions of our research. Hence, the article is structured into five distinct sections, each serving a specific purpose. The introductory segment, following the IMRaD framework, sets the stage by offering an overview and contextual background. In this section, we emphasize gaps in existing literature, laying the groundwork for further inquiry, and delineating the research questions to be addressed. The subsequent section explores web discourse analysis techniques, augmenting the conventional IMRaD structure. It offers a comprehensive exploration of online discourse analysis methodologies, encompassing a detailed examination of both machine learning and lexicon-based techniques. By elucidating these methods, we provide readers with a profound

understanding of the analytical tools available. The methodology section, following the IMRaD framework, elucidates the step-by-step process involved in conducting online discourse analysis. This serves as a practical guide for researchers, offering a clear roadmap for their own investigations. The Results and Discussion section, adhering to the IMRaD format, introduces an ODA framework specifically tailored for deciphering hybrid threats. In this framework, we not only delineate its components but also engage in a critical discussion of the challenges it may face and the opportunities it presents. Finally, the concluding section supplements the IMRaD structure by discussing the significant theoretical and political contributions of our study. While acknowledging the limitations of our research, we offer insightful recommendations for future investigations, ensuring the continuation of scholarly discourse and exploration in this field.

## 2. Literature review

### 2.1. Web discourse analytics techniques

ODA is a well-established field that investigates language and communication trends in digital interactions. This discipline encompasses dialogues, debates, and exchanges, addressing the increase in online communication. ODA covers bias analysis in news, politics, and novel domains like space security. Leveraging Natural Language Processing (NLP) and ML, ODA dissects spoken and written content. This section briefly explores ML's role in Sentiment Analysis (SA) and the lexicon-based approach.

### 2.2. Analysis of specific online content

ODA stands as a well-established field of inquiry, dedicated to the examination and comprehension of language nuances and communication trends permeating diverse forms of online interactions. This discipline encompasses an in-depth analysis of dialogues, debates, and exchanges unfolding across the digital landscape, spanning social media platforms, discussion forums, comment sections, and beyond. This imperative arose in response to the exponential surge in digital communication, which necessitated a profound exploration of its multifaceted dimensions.

A prominent avenue of inquiry within ODA revolves around the examination of biased and prejudiced expressions within online news platforms [6], such as gender issues [7], the complex domain of politics [8], among other pertinent topics. In practical terms, discourse analysis involves a comprehensive study of language within its broader social context. In this light, discourse analysts exhibit equal interest in dissecting spoken discourse and scrutinizing written discourse [9]. Considering the technological advances in recent years, NLP became a growing field of computational science that aims to model natural human language [10]. Combined with the advancements in ML, which entails learning patterns from vast datasets, NLP has bestowed upon us practical capabilities, including the automation of language analysis.

### 2.3. Machine Learning (ML) technique

ML is a subset of Artificial Intelligence (AI) [11] and encompasses the formulation and refinement of algorithms and models [12]. These constructs empower devices to derive insights and reach determinations autonomously, all while circumventing the need for explicit programming. In the context of ODA, ML identifies patterns, prevalent trends, and sentiments amidst copious amounts of online content. Foremost among these techniques lies SA [13], a prevalent ML application within ODA. SA orchestrates the task of ascertaining the prevailing sentiment encapsulated within textual constructs like online comments. To achieve this, SA leverages the

process of training algorithms on labeled datasets, wherein each textual fragment is affixed with a sentiment label—ranging from positive and negative to neutral [14].

Once these algorithms undergo training, these algorithms can analyze new textual data and prediction the inherent sentiment. The SA journey traverses several crucial steps, starting with the preprocessing of text, encompassing activities like tokenization. Following, feature extraction, wherein words are represented into numeric arrays. This process then is marked by the training of a classifier on these annotated datasets, equipping the algorithm to accurately predict sentiments when confronted with novel textual inputs.

## 2.4. Lexicon-based technique

The lexicon-based approach within the ODA distinguishes itself from ML by leveraging predefined dictionaries [15] or lexicons containing words and phrases associated with distinct emotions, sentiments, or topics [16]. Unlike ML, this technique does not necessitate algorithms or extensive training on vast datasets. Instead, it relies on the alignment of words within the text to entries within the lexicon, culminating in the computation of comprehensive sentiment or thematic insights based on these correspondences.

A sentiment lexicon serves as a repository of words or phrases, each annotated with a sentiment classification (positive, negative, neutral). Within lexicon-based SA, the sentiment scores attributed to words detected in the text are combined to derive a sentiment score. While lexicon-based methods exhibit simplicity in implementation and computational efficiency, they are not without constraints. Challenges such as sarcasm, interpreting context-dependent meanings, and accommodating novel terminology absent from the lexicon can impede their effectiveness [17]. In pursuit of enhanced accuracy and the ability to capture subtleties in online content, a synergistic relationship between lexicon-based techniques and ML can be encouraged. This combination can potentially mitigate the limitations of each approach, leading to a more robust and refined SA process, and for that reason, in our conceptual framework, we include ML as optional.

Fig 1 is adapted from Che et al. [18] and Taboada [19], and presents a summary of this section.

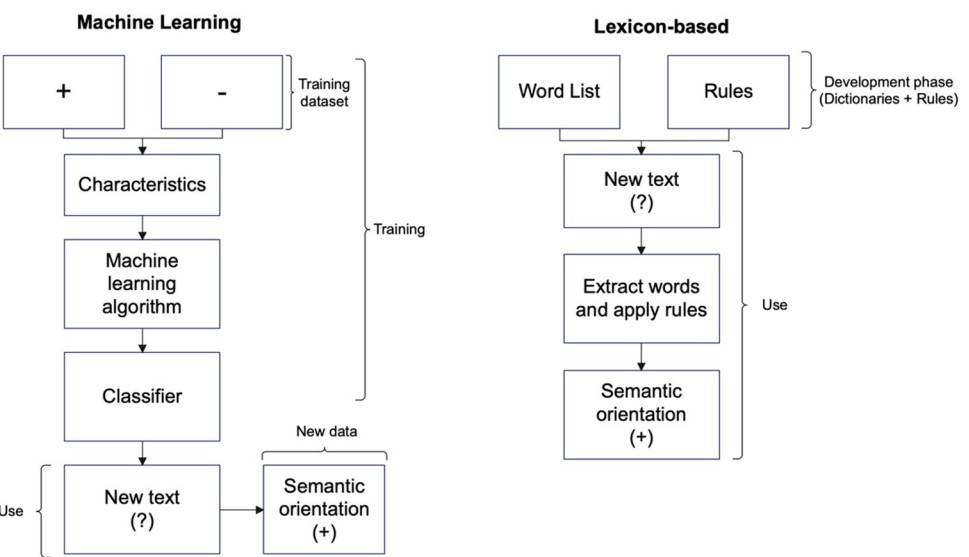

**Fig 1. Machine learning and lexicon-based approaches.** Adapted from [18, 19].

## 3. Methodology

The 8-step framework presented herein offers a systematic approach to addressing the complex challenge of hybrid threats within the context of space-related technologies. To elaborate on the 8-step framework, a conventional literature review was undertaken. The selection of information sources focused on enhancing credibility, thereby encompassing the databases Web of Science (WoS) and Elsevier Scopus, renowned for their repository of peer-reviewed manuscripts. Supplementary inclusion of select documents from Google Scholar was occasionally incorporated to ensure comprehensive coverage across diverse scholarly contributions.

### 3.1. Step 1 –Space communities´ detection

The initial stage entails the systematic collection of data from adequate online sources, encompassing diverse platforms such as discussion forums and social media hubs that are centered around discourse pertaining to the European space domain. This assortment includes online platforms like forums frequented by space enthusiasts and specialized subreddits dedicated to space-related topics. Following this data aggregation, the procedure of web scraping is initiated by Karthikeyan et al. [20]. This technique involves the automated extraction of information from websites, often facilitated by tools such as Beautiful Soup [21]. However, it is essential to recognize the presence of a substantial and impactful reservoir of digital information residing on the Deep Web [22] and Dark Web [23]. The Dark Web introduces a challenge due to its encrypted segments, which lie beyond the scope of traditional search engine indexing mechanisms [24]. Consequently, these encrypted portions remain unaccounted for in search engine result pages, rendering them beyond reach for users' typical inquiries. Effectively accessing, processing, and disseminating content from these obscured segments requires the adoption of specialized protocols or the exploration of alternative sources. Nevertheless, it is noteworthy that resorting to secondary sources, as highlighted by the work of Bieteniece et al. [4], carries the potential to introduce biases, thereby potentially leading to the curtailment or even alteration of the original dataset.

### 3.2. Step 2 –Text preprocessing

Text preprocessing constitutes a pivotal preliminary phase preceding NLP and text analysis [25]. This phase encompasses the refinement and metamorphosis of raw textual data into a structure suitable for subsequent analysis and modeling [26]. The principal objective underpinning text preprocessing is the removal of extraneous noise, and irrelevant information, and the establishment of textual uniformity. Illustrative instances of these preprocessing operations encompass the following: 1) Lowercasing: this operation entails the conversion of all text into lowercase [27]. This process ensures the equitable treatment of words irrespective of their capitalization; 2) Punctuation and Special Character Removal: this operation involves the excision of punctuation marks, symbols, and non-alphanumeric characters that lack substantive contribution to the core semantic essence of the text [28]. Examples encompass punctuation marks such as periods, commas, exclamation marks, and hashtags; 3) Stop word Elimination: stop words, typified by "and," "the," "is," "in," and their ilk, are generally devoid of semantic gravitas [29]. Oftentimes, they are expunged to mitigate data dimensionality and enhance processing efficiency.

Nevertheless, a persistent scholarly discourse revolves around the efficacy of excluding stop words, particularly within specialized domains [30]. It is noteworthy that in specific contexts, such as SA, retention of stop-words may yield utility. We have incorporated the technique into our analysis (8-Step ODS), despite its current absence of empirical validation in the context of space affairs. This issue will be tested further in future research while conducting the empirical

analysis; 4) Tokenization: tokenization entails the dissection of text into constituent words or phrases, known as tokens [31]. Given that most NLP endeavors hinge upon the manipulation of individual words as fundamental analytical units, tokenization is integral. This process involves segmenting text at whitespace junctures, with complexities arising in cases involving contractions (e.g., "can't"), hyphenated words, or linguistic subtleties; 5) Stemming and Lemmatization: these techniques are wielded to distill words to their basal or root forms [28]. Stemming entails the excision of prefixes or suffixes to derive the root word (e.g., "running" into "run"). In contrast, lemmatization goes further, transmuting words into their canonical manifestations (e.g., "better" into "good"). These techniques engender word standardization, diminishing lexical variations and potentially heightening the precision of text analysis [32].

Moreover, other pertinent considerations encompass contraction handling, HTML tag, and URL management [33], spell-checking, and correction [34]. The cumulative effect of these measures culminates in text data that is cleansed, standardized, and tokenized, constituting a state of preparedness for an assortment of NLP endeavors, including SA and text classification. The extent of requisite preprocessing hinges upon the attributes of the data, notably its inherent composition–space related, and the specific analytic aims, characterized in this context–identification of hybrid threats.

### 3.3. Step 3 –Keyword extraction

Subsequently, the classification of hybrid threats and security quandaries inherent to the space defense arena assumes paramount importance. This segmentation encompasses diverse facets, encompassing terminologies germane to cyber perils, disinformation campaigns, technological espionage, orbital debris concerns, satellite jamming, data breaches, and more. It is needed to compile an exhaustive collection of pertinent keywords tailored to each category of threat. Equally deserving of attention are the acronyms and abbreviations entrenched in this domain, exemplified by SST (Spatial Surveillance and Tracking), SSA (Space Situational Awareness), and GNSS (Global Navigation Satellite System), among others.

Two salient considerations warrant diligent attention. Foremost, the imperative to sustain the database's currency by accommodating novel terms, given the rapid evolution within the precincts of space defense and security [36]. This endeavor mandates consistent collaboration with EU defense and space security authorities, engendering a process that validates and fine-tunes the array of keywords. Secondly, multilingual terms warrant inclusion, cognizant of the European Union's encompassing of linguistically diverse member states. This initiative not only amplifies the purview of relevant content capture but also acknowledges the global provenance of hybrid threats.

Upon curating the selected keyword repertoire, counter-hybrid threat experts can seamlessly integrate the repertoires with state-of-the-art keyword extraction techniques facilitated by Python libraries such as Natural Language Toolkit (NLTK), spaCy, or sophisticated text mining platforms. This synergy catalyzes the discernment and analysis of hybrid threats and security quandaries permeating online textual content germane to EU space defense and security.

### 3.4. Step 4 –Natural Language Processing (NLP)

Text analysis encompasses various techniques, with SA being a prominent one. SA as an NLP approach involves dissecting text excerpts to unveil the emotional tone or sentiment they convey [37]. The objective is to categorize emotions spanning the spectrum from positive to negative, encompassing diverse levels of intensity such as strong and weak emotions [38]. Our

initial phase focuses on discerning whether the discourse contains content related to hybrid threats. Subsequently, we explore the type and intensity of the sentiment expressed.

In a landscape dominated by hybrid threats, SA proves valuable for deciphering the emotional tones in diverse forms of communication, ranging from news feeds, social media posts, to specific online forums [39]. Given that hybrid threats often involve tactics designed to manipulate public sentiment or foster confusion [40], employing SA becomes essential. This analytical approach allows experts to spot adverse or deceptive sentiments that might indicate efforts to provoke discord, spread falsehoods, or incite turmoil. By doing so, security specialists and policymakers gain deeper insights into the motivations and strategies underlying hybrid threats, facilitating the implementation of effective countermeasures.

Named Entity Recognition (NER), another integral NLP technique revolves around identifying and classifying specific entities or terms within a text [41]. These entities encompass individuals' names, organizational names, geographical locations, dates, and more [42]. NER algorithms leverage diverse methods, including ML and rule-based models, to accurately pinpoint and categorize these entities [43]. In the sphere of hybrid threat detection, NER techniques are pivotal for unearthing pieces of information that may hint at the presence of hybrid threat activities. For instance, in a discourse centering around potential hybrid threats, NER could swiftly identify mentioned country names, thereby shedding light on the geographic extent of the threat. Furthermore, NER can unveil references to organizations or groups implicated in dubious actions. Notably, NER is adept at flagging specific terms or keywords related to hybrid tactics, such as propaganda, cyber-attack, or disinformation, which frequently accompany hybrid threats. Applying NER techniques to textual content empowers analysts to automatically distill and categorize crucial insights, simplifying the detection and response to hybrid threats in a timely manner. These techniques substantially augment situational awareness and the capacity to derive actionable intelligence from extensive unstructured data. Overall, SA and NER stand as potent pillars within the domain of NLP, equipped to identify, and mitigate hybrid threats. SA facilitates the comprehension of emotional undercurrents and underlying intentions in communications, whereas NER aids in recognizing entities and terminology linked with hybrid threats. Collectively, these techniques form a holistic strategy for hybrid threat detection and response, a strategy we are poised to explore further.

### 3.5. Step 5 –Hybrid threat detection

The foremost challenge in this undertaking revolves around identifying potential threats. To address this challenge effectively, a more profound exploration is imperative, extending beyond the surface-level assessment (SA). In this phase, we can harness two methods: Network Analysis (NA) and Contextual Analysis (CA). These methods have demonstrated their efficacy across various domains.

Commencing with NA, this encompasses a set of integrated techniques designed to delineate interactions between actors and to analyze the social structures that emerge from the recurrence of these relations [44]. This entails a study of connections between diverse entities, ranging from individuals to organizations. The representation of these connections often takes the form of network graphs, where nodes symbolize entities and edges signify the relationships binding them [45]. NA serves to comprehend the dynamics of information dissemination, the propagation of influence, and the flow of resources within a given system. Regarding hybrid threats—characterized by the strategic fusion of both conventional and unconventional tactics to attain specific goals—NA emerges as a relevant tool. The nature of these threats frequently demands orchestrated efforts and partnerships between disparate players, spanning from governmental bodies to non-state actors [46], operating across diverse spheres such as cyber

operations, propaganda, and military operations. Through the lens of NA, these hybrid threat landscapes can be dissected, enabling analysts to unearth pivotal participants, comprehend their functions, delineate communication pathways, and pinpoint potential vulnerabilities. This information can help in devising effective strategies to combat or mitigate these threats.

CA entails the examination of data within its wider framework to identify patterns, trends, and irregularities [47]. This method explores not solely the data itself, but also the circumstances encompassing it, thus endowing analysts with more profound insights. Particularly in the domain of hybrid threats, CA serves to pinpoint atypical or abnormal behaviors. For instance, a sudden surge in online activity centered around a specific space subject might indicate a synchronized endeavor, such as a disinformation campaign or a cyber assault. Through the analysis of this activity's context—including timing, sources, and content—analysts can better understand the motivations and potential repercussions.

In essence, both NA and CA assume pivotal roles in comprehending and mitigating hybrid threats. NA centers on mapping relationships and interconnections among diverse entities involved in hybrid threats, thereby enlightening the structure and dynamics of these challenges. On the other hand, CA examines data within a broader context, unearthing patterns, and anomalies, thereby facilitating the detection of coordinated undertakings or emergent trends related to hybrid threats. By synergistically employing these methods, analysts can gain a more comprehensive understanding of hybrid threats, thereby enabling the formulation of efficacious countermeasures and responses essential for safeguarding security and stability.

## 3.6. Step 6 –ML classification (optional)

ML classification constitutes a specialized domain within the broader sphere of AI [48], where the focus lies in training a model to systematically sort and allocate data into predetermined classes or categories. However, the implementation of a machine-learning classification system is far from straightforward. The existing literature presents divergent viewpoints on numerous crucial aspects, adding complexity to the decision-making process [49]. When placed within the context of discerning patterns intertwined with hybrid threats, the ultimate aspiration entails constructing a model with the proficiency to discern, with precision, between diverse classifications of hybrid threats. This discernment hinges upon the model's capacity to decipher a multitude of distinctive attributes or recurring trends embedded within the data. The terminology "hybrid threats" pertains and multi-dimensional security predicaments, orchestrating the fusion of disparate stratagems and constituent components [50]. The endeavor to fashion a ML model tailored for the identification of patterns affiliated with hybrid threats assumes the role of a noteworthy instrument at the disposal of security analysts and decision-makers. By affording them an augmented comprehension of the intricacies at play, this model bolsters their capacity to not only comprehend but also effectively counteract these nuanced threats.

## 3.7. Step 7 –Real-time monitoring and alerting mechanism

Enhancing the identification of hybrid threats calls for the establishment of a dynamic real-time monitoring framework. This involves deploying automated scripts that operate continuously, overseeing and analyzing the evolution of diverse information sources. The goal is to maintain an uninterrupted stream of data input, ensuring awareness of emerging information or updates. Enabling real-time monitoring affords ample time for the analysis of preemptive measures and effective counteractions against hybrid threats. Real-time monitoring has already found application in the research community, as exemplified by Jachim et al. [51]. In their study, the authors introduce TrollHunter2020—a real-time detection engine—leveraged

to track trolling narratives on Twitter during and post the 2020 US elections. These trolling narratives emerged as alternative explanations for divisive occurrences, intended to fuel information manipulation or evoke emotional reactions. The identification of trolling narratives proved pivotal in upholding constructive conversations on Twitter and curbing the dissemination of misinformation. The process of detecting such content traditionally consumed a considerable amount of time and data, a luxury often not available in the fast-paced landscape of rapidly evolving elections marked by high stakes. To surmount this challenge, the authors engineered TrollHunter2020 to proactively hunt for trolling narratives. They achieved this by analyzing numerous Twitter trending topics and hashtags associated with candidate debates, election nights, and the aftermath of elections. Through correspondence analysis, TrollHunter2020 unveiled significant relationships between nouns and verbs employed in the construction of trolling narratives, pinpointing their emergence in real-time Twitter conversations. The outcomes of the study underscore the efficacy of TrollHunter2020 in promptly capturing nascent trolling narratives during the initial phases of polarizing events. The authors explored the implications of TrollHunter2020's utility, highlighting its potential for the early detection of information manipulation and trolling. Moreover, they also explored the tool's role in facilitating more controlled discussions around polarizing subjects on the platform.

Furthermore, the implementation of alert mechanisms serves as an additional layer of vigilance. These mechanisms can be developed to efficiently notify analysts of suspicious patterns. This task can be achieved through swift email notifications, messaging applications, or the generation of comprehensive reports that outline the identified activities. Tailoring the alerting mechanism to the specific exigencies of the organization or context is essential, allowing for the prioritization of alerts based on their severity and relevance. Through the integration of advanced analytics and ML algorithms, the system can draw insights from historical data and dynamically adjust its alert thresholds over time. This iterative refinement process enhances the system's precision while diminishing the occurrence of false positives. This proactive approach not only facilitates swift responses to potential threats but also nurtures the maturation of the system's detection prowess as it evolves. By coupling real-time monitoring with adaptive alerting, organizations can improve their hybrid threat identification capabilities, ensuring a comprehensive and agile defense posture.

## 3.8. Step 8 –Refinement and iteration

Managing space-related technologies and hybrid threats represents a notably challenging task, given its requirement for consistently evolving technical assessment. As hybrid threats undergo changes, so must the response strategies. To accomplish this, seeking guidance from specialists across various domains such as cybersecurity, international relations, and intelligence becomes essential for obtaining comprehensive and suitable insights, including potential countermeasures. While each of the 8-steps outlined in this article holds importance, their effectiveness heavily relies on appropriate counsel. Without it, any technological progress risks faltering due to a lack of alignment. Consequently, prior to embarking on technological development, establishing a panel of experts, and conducting surveys to gather insights becomes vital. Nonetheless, this process extends beyond this initial phase, necessitating ongoing and cyclic monitoring and applying countermeasures against hybrid threats. Concurrently, the iterative nature of scientific research contributes by equipping experts with novel and relevant findings. In conclusion, effectively addressing the convergence of space-related technologies and hybrid threats demands a methodical approach involving consultative processes, sustained vigilance, and the reciprocal relationship between scholarly exploration and applied expertise.

## 4. Results and discussion

This section presents the ODA framework, along with the limitations and challenges associated with current hybrid threat detection methods in the context of space operations. Furthermore, it is also discussed how advanced ODA techniques can be leveraged to overcome these limitations and provide a more comprehensive understanding of emerging threats.

### 4.1. Conceptual framework–ODA to decode hybrid threats

In this first section, we present the ODA framework designed to assess hybrid threats to EU space defense and security (Fig 2). The process initiates with the selection of the appropriate programming language, source code editor, and operating system. For this instance, Python is chosen as the programming language, particularly for empirical validation, together with the

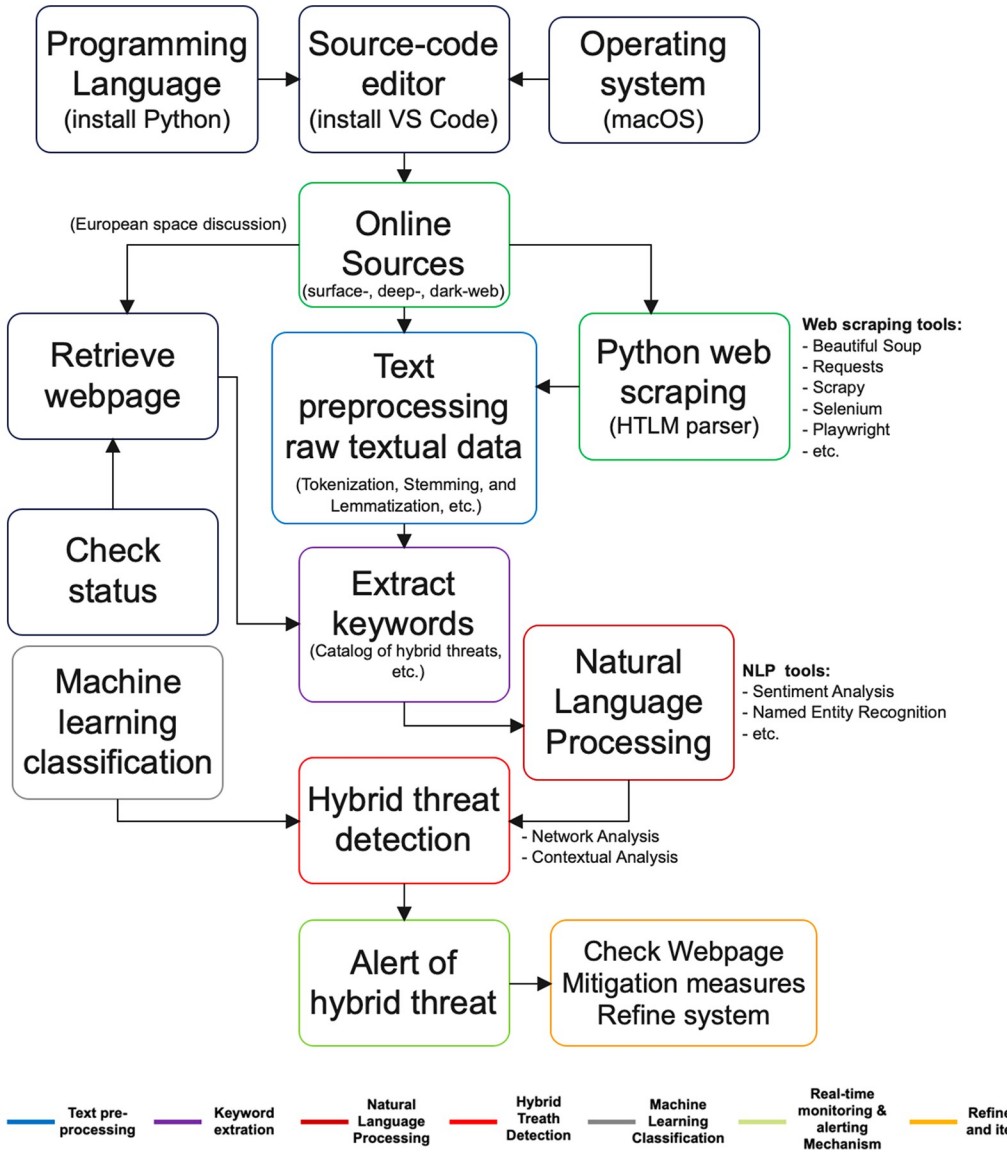

**Fig 2. Online Discourse Analysis (ODA) framework for deciphering hybrid threats to the EU space defense and security.**

Visual Studio (VS) Code editor in the macOS operating system. Subsequently, it is necessary to identify the communities linked to the European spatial domain.

Data collection involves the application of Python web scraping techniques. This necessitates the installation of a package capable of parsing HTML and XML documents, such as "Beautiful Soup." Post-collection, text preprocessing commences, involving the refinement and conversion of unstructured text data into a suitable format for analysis. Keyword extraction is conducted using a catalog of predefined keywords related to hybrid threats in the European space domain, which correlates with the target webpage. Natural Language Toolkit (NLTK), spaCy, or similar text mining platforms are deployed to facilitate the extraction of keywords from the text.

Following the confirmation of URL status and content extraction, two key analyses are performed–SA and NER. These analyses contribute to a more comprehensive understanding of the content. Eventually, the results obtained from the pre-established SA and NER processes are presented. The final stage of analysis encompasses mapping interactions and relationships among entities implicated in hybrid threats through Network Analysis. Additionally, exploration of data patterns and irregularities is executed to identify coordinated efforts, thereby initiating Contextual Analysis. As a subsequent possibility, the development of ML models can be considered for the classification of hybrid threat patterns.

Concluding this framework, a dynamic monitoring structure is implemented via automated scripts. This framework ensures the continuous evaluation of evolving information sources, coupled with the activation of alert mechanisms. To complete the cycle and recommence the process, specialists from various domains, particularly those well-versed in hybrid threats and space systems/technologies, are invited. This collective input enables the refinement of the entire system, including threat categorization, leading to a more effective and robust analysis process.

## 4.2. Issue 1 –Finite pool of experts possessing specialized technical skills

Assessing hybrid threats to space systems demands a thorough understanding and analysis of an array of challenges. Among these, a significant obstacle pertains to the acquisition of experts equipped with specialized training tailored to the space and hybrid threats contexts [52]. Various illustrative instances emphasize this assertion: The scarcity of solid countermeasures is evident, for example, the legal framework that governs space activities is still evolving [53]. This complexity impedes effective responses to hybrid threats and the determination of actions constituting aggression in space. That is, as the threat itself is usually diffuse, in these cases its identification is more difficult, even for specialists in the field [54].

The complexity of the threat landscape furthers turn the response more difficult. The same to say, proficiently comprehending and countering the broad spectrum of hybrid threats necessitates expertise spanning diverse knowledge domains. This includes specialists to comprehend not only cyber defense but also a grasp of e.g., geopolitical matters and beyond. The interconnectedness of space systems encompasses numerous infrastructures [55]. The repercussions of disruptions to space-based communication and navigation systems, for instance, extend well beyond the confines of the space sector, affecting domains such as transportation, finance, and emergency services [56]. Addressing these limitations and challenges mandates an interdisciplinary approach that combines technical expertise with an ongoing formulation of policies capable of adapting to the continually evolving threat landscape.

## 4.3. Issue 2 –Highly adaptable and skilled opponents

A second crucial concern revolves around the adaptability displayed by adversaries [57]. As space technology advances at a swift pace, potential foes are also poised to devise novel methods for

capitalizing on vulnerabilities within space systems. Staying abreast of the latest technological progress and tailoring countermeasures accordingly presents an ongoing challenge.

Furthermore, a pair of additional dimensions come into play. Firstly, the dual-use nature inherent in civil-oriented space systems [58] can limit attempts to discern between legitimate activities and potential hybrid threats. This introduces a second challenge, the development and upkeep of countermeasures against hybrid threats can be resource intensive [59]. Securing the necessary funding and expertise for this endeavor can prove demanding, particularly when contending with other imperatives in the dominion of national security.

Lastly, in the context of this article, limited access to data emerges as an additional challenge. Unraveling the source of a hybrid threat is a convoluted undertaking, especially within the domain of space systems. Cyberattacks, disinformation campaigns, and other hybrid maneuvers might be meticulously crafted to obfuscate the origin of the threat, thereby complicating the precise attribution of actions to a specific entity or nation.

## 4.4. Issue 3 – Focus on identifying threats overshadows countermeasures

In the domain of hybrid threats, ODA incorporates elements designed to pinpoint disinformation narratives and their dissemination channels. These narratives are then tackled through a range of strategies, including fact-checking, refutation, and, in certain cases, content suppression. Step 7 of the ODA approach focuses on real-time monitoring and alerting. However, while this step is crucial for threat mitigation, it could benefit from a more explicit emphasis on how ODA can actively mitigate threats, rather than solely identifying and analyzing them.

Presently, European Union policy leans heavily towards threat identification, often at the expense of implementing effective countermeasures [60]. This imbalance persists within the current EU policy framework, posing a significant challenge that must be addressed. Nonetheless, the EU possesses tools such as the Foreign Information Manipulation and Interference (FIMI) Toolbox dedicated to combating hybrid threats [61]. Efforts to realign policy priorities towards proactive threat mitigation remain imperative for tackling the evolving landscape of hybrid threats effectively [62].

At the EU level, the European External Action Service (EEAS) has been actively enhancing its capabilities to detect, assess, and respond to instances of Foreign Information Manipulation and Interference (FIMI). This strategic endeavor aims to bolster the EU's security and defense posture by enabling a more precise and robust counteraction against FIMI threats.

FIMI, often referred to as "disinformation" [61] poses a growing political and security challenge to the EU. The EEAS, particularly through its Strategic Communications, Task Force, and Information Analysis (STRAT.2), has been pivotal in addressing this challenge within the framework of foreign and security policy since 2015 when FIMI first emerged on the EU's policy radar [63]. The EEAS defines FIMI as " a pattern of behavior that threatens or has the potential to negatively impact values, procedures, and political processes. Such activity is manipulative and conducted in an intentional and coordinated manner. Actors of such activity can be state or non-state actors, including their proxies inside and outside of their own territory" [61]. In practical terms, FIMI exacerbates polarization and discord within the EU. In discussing the conceptualization of FIMI, Fridman [64] underscores that while the notion of hybrid threats primarily arises from political-military discussions, the FIMI framework itself is grounded entirely in civilian origins. It was initially introduced in 2021 by the EEAS as a response to the EU Commission's 2020 initiative calling for enhanced conceptual frameworks concerning disinformation. Embraced within various strategic documents, including the 2022 Strategic Compass for Security and Defense, this concept has emerged as a significant term within the professional dialogue and in shaping EU policies.

The EEAS approach integrates policy formulation, methodological frameworks, analysis, and responsive measures to counter FIMI. To this end, the EEAS has established a structured framework consisting of [61]: (1) the Rapid Alert System (RAS) on disinformation for facilitating coordinated actions with other EU institutions and Member States; (2) a comprehensive methodology for systematically gathering evidence from FIMI incidents and disseminating it through the Information Sharing and Analysis Center (FIMI ISAC); and (3) the development of the EU toolbox to combat FIMI (FIMI toolbox). While the first two components focus on early warning and analysis, the FIMI toolbox is designed as a resource for implementing countermeasures. However, it is important to note that the execution of the ODA model primarily concentrates on the initial phases of alerting and analysis. Countermeasures, particularly Tactics, Techniques, and Procedures (TTPs), have been devised but are not publicly shared to avoid jeopardizing EEAS operations–this can potentially represent a limitation of our model. The implementation of countermeasures is slightly addressed in Step 8—Refinement and Iteration. However, due to the need for confidentiality regarding Tactics, Techniques, and Procedures (TTPs), they are not extensively elaborated upon.

## 4.5. Issue 4 –Scholarly literature lags behind on ODA and EU space security hybrid threats

According to the European Center of Excellence for Countering Hybrid Threats (Hybrid CoE), hybrid threats encompass harmful activities meticulously planned and executed with malicious intent [65]. Their primary aim is to destabilize a target, be it a state or an institution, employing a diverse array of tactics, often in combination. The Hybrid CoE emphasizes that these threats are orchestrated by both states and authoritarian regimes, as well as non-state actors (NSAs) frequently acting as proxies for authoritarian entities (e.g., Wagner Group). Notable examples of hybrid threat actors include the Russian Federation, the People's Republic of China (PRC), and Iran [65]. Tactics employed encompass information manipulation [50], cyber-attacks [66], economic coercion [67], covert political maneuvering, coercive diplomacy, and the threat of military force [65]. Hybrid threats span a broad spectrum of harmful activities with diverse objectives, ranging from influence and interference operations to full-fledged hybrid warfare. They offer a cost-effective means of achieving objectives while disrupting the processes and institutions of democratically governed states. Hybrid threat actors strive to evade accountability and countermeasures, crafting their hostile activities to be elusive and challenging to defend against. Their strategies are carefully calibrated to operate beneath the threshold that would constitute or be perceived as an act of war against the targeted state. Despite their elusive nature, we will now outline some activities that have adversely impacted and undermine the European space programs. In that regard, ODA serves as a pivotal tool in identifying various detrimental activities, including (1) Disinformation Campaigns, (2) Sociopolitical Manipulation, and (3) Misinformation on Satellite Activities.

Disinformation Campaigns, firstly, endeavor to erode public confidence and support for space initiatives by disseminating false narratives regarding cost-effectiveness. Such actions pose a tangible threat to the EU's capacity to uphold and enhance security-sensitive satellite infrastructure. For instance, the Kremlin's disinformation endeavors extend beyond space programs, portraying the West as "decadent" (zagnivayushchii Zapad) [68]. Secondly, these campaigns aim to sway political decisions, allowing foreign entities to influence policymakers' perspectives and choices regarding satellite endeavors or collaborations. In that regard, the PRC has significantly expanded the dissemination of biased content, leveraging investments in foreign media outlets to sway political elites and journalists [69]. An exemplification of such activity is the investment in digital television services across Africa and satellite networks,

which can influence potential partnerships with these states and, to a certain extent, affect the conduct of Common Security and Defence Policy (CSDP) missions in these regions [69]. Furthermore, these campaigns strive to sow discord among EU member states by exploiting disparities in national interests or priorities, thereby obstructing collaborative endeavors geared towards bolstering satellite security and interoperability, thus weakening the EU's unified response to security challenges.

Socio-political Manipulation tactics are employed to manipulate dynamics within the EU, utilizing divisive narratives to exacerbate existing social and political tensions. By leveraging online platforms, actors amplify sensitive issues like resource allocation, technological dependencies, or sovereignty concerns, fostering divisions among member states and impeding cooperation initiatives. While like disinformation campaigns, these manipulative efforts run deeper, aiming to exacerbate potentially volatile situations. Another facet involves undermining collaborative endeavors, fostering animosity between member states already harboring suspicions toward one another or EU partner states (including in CSDP missions). Such activities may lead to disparities in joint research projects or mechanisms for sharing critical information necessary to effectively address common satellite security challenges. An illustrative example is the discernible action taken by the People's Republic of China (PRC), resorting to legal measures in democratic societies to silence critics. PRC individuals and organizations have initiated defamation or legal actions against academics within the EU, ultimately impeding the ability to respond to crises [69]. The online manipulation campaign seeks to disrupt crisis response mechanisms reliant on satellite technologies, directing satellite activity away from urgent areas and isolating CSDP missions from humanitarian crises, natural disasters, or security emergencies. Notably, SATCEN's heavy reliance on commercial satellites renders it susceptible to exploitation, with actors capitalizing on the distrust between SATCEN and private/commercial satellite companies that provide services to it [70].

Misinformation campaigns targeting Satellite Activities aim to disrupt European space agencies' operations. This includes spreading false alerts or rumors regarding satellite malfunctions or unauthorized access attempts, leading to public apprehension and potential disruptions in operations. Despite reports from European agencies, the majority either remain unaware of these activities or lack effective mitigation strategies, often shrouded in secrecy.

In conclusion, the utilization of ODA is paramount in uncovering and countering these nefarious activities, safeguarding the integrity and efficacy of EU space programs while bolstering collective resilience against external threats.

## 5. Conclusion

The conclusion is organized into four distinct sections, each with a specific purpose. The initial parts underscore the theoretical and practical contributions concerning space and hybrid threats. Following that, we address research limitations and propose new directions. Lastly, we provide recommendations for future progress, inspiring colleagues to explore this path further.

### 5.1. Theoretical and political contributions

Regarding the theoretical contributions, our article introduces a novel conceptual perspective. It presents an innovative ODA framework within defense and space security, specifically concerning hybrid threats. While the core concept of ODA is not novel, its application in this context is unique and groundbreaking.

The conceptual framework (Fig 2) integrates a suite of tools, including web scraping, NLP, SA, NER, NA, and CA. The systematic employment of these tools allows for the methodical

identification and examination of hybrid threats within online discussions. What truly differentiates this framework are the following key factors:

Comprehensive toolset for dynamic complexity–hybrid threats are notorious for their swift evolution. To address this challenge, our framework employs a diverse array of tools. It recognizes the need to engage experts to continually update the threat catalog, ensuring that the approach remains current and effective; 2) Multidimensional analysis of complexity–the complexity of hybrid threats is dissected from two angles. Firstly, the framework employs network analysis tools to decipher hybrid threat networks. Secondly, contextual analysis tools offer a broader perspective, enhancing the understanding of this complex phenomenon; 3) Simplified Implementation through modular projects–although the framework might appear straightforward, it necessitates a phased approach. By breaking down the implementation into several projects, it becomes feasible to integrate diverse techniques. This contributes to a more nuanced comprehension of the complexity between hybrid threats and online discourse; lastly, 4) Real-time surveillance and alerting mechanism–recognizing the urgency of continuous monitoring and timely responses, the framework incorporates a real-time monitoring and alerting mechanism. This is vital in the face of rapidly evolving threat landscapes, particularly when safeguarding highly valuable space systems.

In conclusion, while seemingly straightforward, this framework represents a complex amalgamation of methods. Its applicability hinges on its division into distinct projects, allowing for the integration of a variety of techniques. By adopting this multifaceted approach, the framework enriches our understanding of the intricate interplay between hybrid threats and online discourse, especially when applied to the protection of valuable space systems.

This article also presents a range of contributions concerning political involvement. These contributions can be categorized as follows:

Policy-Driven threat mitigation–aligned with the political need to increase security and defense against hybrid threats in the EU Space, our article introduces new insights by providing a structured approach to threat analysis. This supports policymakers in devising effective strategies to combat hybrid threats in the context of space security; 2) Enhancing transparency and vigilance–the framework (Fig 2) places a strong emphasis on real-time monitoring and alert mechanisms, thereby enhancing transparency and vigilance in addressing hybrid threats. This aligns with the goal of fostering trust and awareness among stakeholders by swiftly identifying and addressing potential threats; 3) Interdisciplinary collaboration–our article recommends involving specialists from various domains, promoting collaboration, and the exchange of information across sectors. This recommendation is in line with the need to encourage interdisciplinary cooperation to tackle security, ensuring a comprehensive and informed response; 4) Adaptative countermeasures–the framework's iterative and adaptive nature resonates with the need for flexibility in addressing hybrid threats. By incorporating an ML model and continuous refinement, the framework facilitates the development of agile countermeasures that can evolve in tandem with the ever-changing threat landscape; 5) Preemptive measures–Leveraging SA and contextual analysis, the framework aids in the early detection of potential threats and unusual behavior. This aligns with the goal of taking proactive steps to avert the realization or escalation of hybrid threats; 6) Recognizing global implications–this article acknowledges the global ramifications of hybrid threats by considering multilingual terms and diverse linguistic origins. This aligns with the policy aim of fostering international collaboration to address hybrid threats that transcend geographical boundaries; finally, 7) Technological readiness–the article's focus on data pre-processing, advanced NLP techniques, and ML models, which are consistent with the aspiration of maintaining technological readiness to combat evolving hybrid threats. This enhances the European Union's capabilities in space defense and security.

In summary, the article's theoretical and policy contributions revolve around its methodical framework for comprehending, identifying, and responding to hybrid threats in the context of EU space defense and security. Our framework aligns with the strategic priorities of augmenting security measures, cultivating collaboration, and sustaining technological preparedness regarding dynamic hybrid threat scenarios.

## 5.2. Research limitations

This article is not exempt from certain research limitations that warrant attention. We would like to underscore three specific research limitations that, from our perspective, hold significance.

Firstly, a research limitation arises in relation to the conceptual framework. Given the rapid evolution of hybrid threats and the ongoing advancements in AI and computer science [71, 72], it is reasonable to anticipate a corresponding evolution in the literature. Consequently, there will be a need for continuous adaptation to remain current. In essence, this article captures a snapshot of the present reality, which may inevitably lag the cutting-edge developments in a matter of a few years. Secondly, the comprehensiveness of the identified steps and the conceptual framework implies that the envisioned technology necessitates division into multiple distinct projects. For instance, one feasible project can entail the creation of an online monitoring system, akin to the exemplified TrollHunter2020 discussed earlier. Dividing the technology's development into discrete projects is an approach to effectively manage the complexity inherent in the pursuit of such a challenging goal. Lastly, a noteworthy limitation pertains to the continuous enhancement imperative for the software(s) we intend to develop. This imperative is rooted in the requirement to remain aligned with the rapid advancements in space-related technologies and the hybrid threats that emerge in tandem. Continuous refinement of the software(s) is indispensable to ensure its relevance and efficacy in a landscape characterized by constant evolution.

Acknowledging and addressing these research limitations will be pivotal in crafting a robust and adaptive approach that stands resilient against the challenges posed by hybrid threats and technological dynamism in the space domain.

## 5.3. Recommendations for future research

Empirical validation is on the brink of realization through forthcoming scientific research endeavors that involve the development and deployment of cutting-edge software using Phyton programming. Anticipated as an instrumental tool for the future, this software is envisioned to be incorporated within the arsenals of various space agencies across EU member-states. By promptly detecting emerging threats, EU national space agencies and direct responders can collaborate closely with the Commission, namely, the Computer Security Incident Response Team of all EU institutions (CERT-EU), and the European Union Agency for Cybersecurity (ENISA), to execute synchronized responses. This collaborative synergy serves a dual purpose: not only does it enhance the security and defense of EU space assets and interests, but it also erects a resilient stronghold of defense against hybrid threats. Thus, the framework we propose encompasses harmonized processes for identification, analysis, and response, underscoring the dedication of the EU to safeguarding its strategic space domain against the challenges posed by hybrid threats. This framework stands as evidence of the commitment of the EU and its member states to ensure the safety and integrity of their space domain.

Furthermore, the forthcoming investigation necessitates a comprehensive examination of adherence to ethical and legal standards. Particularly within the framework of developing and

implementing software designed to identify hybrid threats against European space, safeguarding all collected data becomes imperative. This data must remain confidential, thwarting any unauthorized access or misuse. Guidance from institutional review boards or ethics committees is often sought to ensure alignment with ethical principles and global best practices.

## Acknowledgments

We express our gratitude to COFAC–Cooperativa de Formação e Animação Cultura CRL for generously funding the Article Processing Charge (APC) and providing valuable support throughout the research process.

## Author Contributions

**Conceptualization:** João Reis.

**Data curation:** João Reis.

**Formal analysis:** João Reis.

**Funding acquisition:** João Reis.

**Investigation:** João Reis.

**Methodology:** João Reis.

**Project administration:** João Reis.

**Resources:** João Reis.

**Software:** João Reis.

**Supervision:** João Reis.

**Validation:** João Reis.

**Visualization:** João Reis.

**Writing – original draft:** João Reis.

**Writing – review & editing:** João Reis.

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
