## [Decision Letter · Decision Letter 0]

25 Jan 2024

PONE-D-23-34933EU Space Security – An 8-Step Online Discourse Analysis to Decode Hybrid ThreatsPLOS ONE

Dear Dr. Reis,

Thank you for submitting your manuscript to PLOS ONE. After careful consideration, we feel that it has merit but does not fully meet PLOS ONE’s publication criteria as it currently stands. Therefore, we invite you to submit a revised version of the manuscript that addresses the points raised during the review process.

We look forward to receiving your revised manuscript.

Kind regards,

Michal Ptaszynski, PhD

Academic Editor

PLOS ONE

Journal Requirements:

Reviewers' comments:

Reviewer's Responses to Questions

**Comments to the Author**

1. Is the manuscript technically sound, and do the data support the conclusions?

Reviewer #1: Yes

Reviewer #2: Partly

2. Has the statistical analysis been performed appropriately and rigorously? 

Reviewer #1: I Don't Know

Reviewer #2: N/A

3. Have the authors made all data underlying the findings in their manuscript fully available?

Reviewer #1: No

Reviewer #2: Yes

4. Is the manuscript presented in an intelligible fashion and written in standard English?

Reviewer #1: Yes

Reviewer #2: Yes

5. Review Comments to the Author

Reviewer #1: The research topic "EU Space Security – An 8-Step Online Discourse Analysis to Decode Hybrid Threats" is very interesting. Below is a description of the situation of the article with its respective recommendations:

1.They present original results with the definition of 8 steps that follow a clear methodology.

2.The experiments carried out using python software are appropriate

3.The conclusions they make regarding "Theoretical and political contributions" are appropriate based on the contribution

4.The writing of the article in English is adequate but should improve by applying the IMRYD methodology

5.It must define the applicable standards for the ethics of experimentation and the integrity of the research, for the article to be accepted

6.Make a description or justification because the data is not available to everyone.

With the above background, it is recommended to accept, complying with the defined suggestions.

Reviewer #2: This paper presents an interesting and comprehensive account of the use of Online Discourse Analysis (ODA) as a means of identifying and countering hybrid threats, ostensibly using EU space security as a case study in which to understand these processes/a field in which this technique can be applied. It is a paper that I believe has a lot of potential, but needs some significant revisions in order to ensure this is the case.

The paper presents two research questions: the first on how ODA can be used to identify, analyse and mitigate hybrid threats, which I feel the paper effectively addresses through its comprehensive discussion and explanations, and I do not have any particular concerns with; the second is what limitations current approaches to hybrid threats have that ODA can address, the answer to which which I feel is much less robust and developed in the paper as currently written. The current approaches do not appear to be explicitly spelled out, but are instead hinted towards or mentioned obliquely, making it difficult to identify what current approaches are used, where there limitations are, and how these new techniques may be more beneficial. For example, when thinking about hybrid threats, it would appear that elements of the ODA approach exist in the current approaches, be they using researchers and experts to identify (for example) disinformation narratives, points of distribution/dissemination, and then challenge them through various mechanisms such as fact-checking, rebuttal, or in some cases, suppression/deprioritisation of content. Step 7 of the ODA approach discusses real time monitoring and alerting, but if one of the key aspects of these processes is threat mitigation, Step 7 may want to focus more on how exactly ODA can be used to mitigate, rather than just identify and analyse hybrid threats.

Another issue I have with the paper is the use of hybrid threats in the context of EU space policy as a framing for the paper. I feel the paper needs to provide much more explanatory detail on what a hybrid threat is, how these are defined and currently approached by the EU (if the EU is the focus), and why space is a field in which assessment of hybrid threats is of relevance. Much of the contemporary focus on hybrid threats concerns the use of techniques such as disinformation combined with attacks on information infrastructures intended to destabilise or delegitimise, while falling short of methods considered conventional in warfare. For this reason, a lot of the EU's policies and actions in this field have been focusing on issues such as 'democracy' (with the EU Democracy Action Plan), electoral integrity, improving regulation of political advertising, and co-regulatory mechanisms including under the Digital Services Act to bring social media platforms into a more effective regulatory space. In the context of space, what are the hybrid threats that are presented? I can see the obvious cyber-security related dimension, but where would the disinformation/discursive element be, that generates the 'hybrid' threat? I feel that the paper's theoretical contribution could be much stronger if it engaged more effectively with the literature on hybrid threats, situating the ODA model into this analysis, and then exploring how it could be effectively used. Section 4, which provides the discussion, does a little of this implicitly, but it needs to be greatly strengthened. It appears to acknowledge that such a system has a strong need for the involvement of experts in these hybrid threats in order for this approach to be effective - where in the ODA model are they brought in? Do they help inform the text reprocessing and keyword extraction? Or are they intended more for the 'mitigate' phase, rather than the identify and analyse phases? I feel the paper needs to be much clearer on this.

In summary, I think this really is a paper with a lot of potential, and a model that has the potential to be supportive in combating the discursive element of hybrid threats - however, it needs more work to make this contribution clearer.

6. PLOS authors have the option to publish the peer review history of their article (what does this mean?). If published, this will include your full peer review and any attached files.

Reviewer #1: **Yes: **Segundo Moisés Toapanta Toapanta

Reviewer #2: No

---

## [Author Response · Author response to Decision Letter 0]

28 Feb 2024

# GENERAL COMMENTS TO REVIEWERS

First, we would like to thank the time you devoted to providing insightful recommendations. We believe that your comments have improved the paper in many ways. We hope you agree that the revised version builds a stronger contribution.

We have taken the revision very seriously. If you feel that the revision has fallen short, we can reformulate accordingly. The following paragraphs deal with the detailed comments you raised.

REVIEWER #1

Reviewer point #1: The research topic "EU Space Security – An 8-Step Online Discourse Analysis to Decode Hybrid Threats" is very interesting. Below is a description of the situation of the article with its respective recommendations: (1) They present original results with the definition of 8 steps that follow a clear methodology; (2) The experiments carried out using Python software are appropriate; (3)The conclusions they make regarding "Theoretical and political contributions" are appropriate based on the contribution; (4)The writing of the article in English is adequate but should improve by applying the IMRYD methodology; (5) It must define the applicable standards for the ethics of experimentation and the integrity of the research, for the article to be accepted; (6) Make a description or justification because the data is not available to everyone.

With the above background, it is recommended to accept, complying with the defined suggestions.

Author response #1: First, we would like to express our gratitude for your insightful comments, which have greatly contributed to the refinement of our work. In response to your suggestions, several key adjustments have been implemented. We have enhanced the clarity of the article organization, aligning it more closely with the IMRAD (Introduction, Methods, Results, and Discussion) structure. Furthermore, we have integrated arguments on ethical considerations, emphasizing their significance not only in the current study but also in future research endeavors. Additionally, recognizing the importance of transparency, we have provided a rationale for the absence of extensive data, given the nature of our investigation which primarily focuses on the design of a threat detection model. Should you require further clarification on any of the above aspects or wish to explore other issues of our work, please feel free to do so. Once again, thank you for your support.

REVIEWER #2

Reviewer point #1: This paper presents an interesting and comprehensive account of the use of Online Discourse Analysis (ODA) as a means of identifying and countering hybrid threats, ostensibly using EU space security as a case study in which to understand these processes/a field in which this technique can be applied. It is a paper that I believe has a lot of potential, but needs some significant revisions in order to ensure this is the case.

Author response #1: Thank you for your comment.

Reviewer point #2: The paper presents two research questions – the first on how ODA can be used to identify, analyse and mitigate hybrid threats, which I feel the paper effectively addresses through its comprehensive discussion and explanations, and I do not have any particular concerns with; the second is what limitations current approaches to hybrid threats have that ODA can address, the answer to which I feel is much less robust and developed in the paper as currently written. The current approaches do not appear to be explicitly spelled out, but are instead hinted towards or mentioned obliquely, making it difficult to identify what current approaches are used, where there limitations are, and how these new techniques may be more beneficial. For example, when thinking about hybrid threats, it would appear that elements of the ODA approach exist in the current approaches, be they using researchers and experts to identify (for example) disinformation narratives, points of distribution/dissemination, and then challenge them through various mechanisms such as fact-checking, rebuttal, or in some cases, suppression/deprioritisation of content. Step 7 of the ODA approach discusses real time monitoring and alerting, but if one of the key aspects of these processes is threat mitigation, Step 7 may want to focus more on how exactly ODA can be used to mitigate, rather than just identify and analyse hybrid threats.

Author response #2: Your point is very well taken. Indeed, within the domain of hybrid threats, there exist elements of ODA aimed at pinpointing disinformation narratives and their dissemination channels. These narratives are then countered through various means, including fact-checking, refutation, and, in certain instances, content suppression or deprioritization. Step 7 of the ODA approach revolves around real-time monitoring and alerts, closely linked to Step 8. The European Union's policy has predominantly prioritized threat identification over effective countermeasures, a reality that persists within the current model and remains a challenge to overcome. Nevertheless, the EU possesses a few tools actively engaged in combating hybrid threats, serving as key points within this framework. To clarify this aspect further, we have provided a more detailed explanation of how ODA facilitates collaboration with experts tasked with formulating strategies to counter hybrid threats (please see section 4.4.). Thank you very much for your comments.

Reviewer point #3: Another issue I have with the paper is the use of hybrid threats in the context of EU space policy as a framing for the paper. I feel the paper needs to provide much more explanatory detail on what a hybrid threat is, how these are defined and currently approached by the EU (if the EU is the focus), and why space is a field in which assessment of hybrid threats is of relevance. Much of the contemporary focus on hybrid threats concerns the use of techniques such as disinformation combined with attacks on information infrastructures intended to destabilise or delegitimise, while falling short of methods considered conventional in warfare. For this reason, a lot of the EU's policies and actions in this field have been focusing on issues such as 'democracy' (with the EU Democracy Action Plan), electoral integrity, improving regulation of political advertising, and co-regulatory mechanisms including under the Digital Services Act to bring social media platforms into a more effective regulatory space. In the context of space, what are the hybrid threats that are presented? I can see the obvious cyber-security related dimension, but where would the disinformation/discursive element be, that generates the 'hybrid' threat? I feel that the paper's theoretical contribution could be much stronger if it engaged more effectively with the literature on hybrid threats, situating the ODA model into this analysis, and then exploring how it could be effectively used. Section 4, which provides the discussion, does a little of this implicitly, but it needs to be greatly strengthened. It appears to acknowledge that such a system has a strong need for the involvement of experts in these hybrid threats in order for this approach to be effective - where in the ODA model are they brought in? Do they help inform the text reprocessing and keyword extraction? Or are they intended more for the 'mitigate' phase, rather than the identify and analyse phases? I feel the paper needs to be much clearer on this.

Author response #3: After reading your comment, we too recognize the significance of bridging literature to our discourse. Though our focus was primarily on technical features, your insight highlights the value of exploring literature in our discourse. In our review, we have provided a comprehensive elucidation of hybrid threats in space, integrating the ODA model into our analysis and delineating its practical application (please see section 4.5.). We hope this version is a considerable improvement over the previous one. Should you require further clarification on any of the above aspects or wish to explore other issues of our work, please feel free to do so.

Reviewer point #4: In summary, I think this really is a paper with a lot of potential, and a model that has the potential to be supportive in combating the discursive element of hybrid threats - however, it needs more work to make this contribution clearer.

Author response #4: We deeply appreciate your constructive feedback, which we believe has significantly enhanced the article. Moreover, we are thankful for the opportunity you have provided us to strengthen the text, making it more impactful and empowering for our community as we progress in this crucial matter.

---

## [Decision Letter · Decision Letter 1]

7 Mar 2024

PONE-D-23-34933R1EU Space Security – An 8-Step Online Discourse Analysis to Decode Hybrid ThreatsPLOS ONE

Dear Dr. Reis,

Thank you for submitting your manuscript to PLOS ONE. After careful consideration, we feel that it has merit but does not fully meet PLOS ONE’s publication criteria as it currently stands. Therefore, we invite you to submit a revised version of the manuscript that addresses the points raised during the review process.

We look forward to receiving your revised manuscript.

Kind regards,

Michal Ptaszynski, PhD

Academic Editor

PLOS ONE

Journal Requirements:

Reviewers' comments:

Reviewer's Responses to Questions

**Comments to the Author**

1. If the authors have adequately addressed your comments raised in a previous round of review and you feel that this manuscript is now acceptable for publication, you may indicate that here to bypass the “Comments to the Author” section, enter your conflict of interest statement in the “Confidential to Editor” section, and submit your "Accept" recommendation.

Reviewer #1: All comments have been addressed

Reviewer #2: All comments have been addressed

2. Is the manuscript technically sound, and do the data support the conclusions?

Reviewer #1: Yes

Reviewer #2: Yes

3. Has the statistical analysis been performed appropriately and rigorously? 

Reviewer #1: Yes

Reviewer #2: N/A

4. Have the authors made all data underlying the findings in their manuscript fully available?

Reviewer #1: Yes

Reviewer #2: Yes

5. Is the manuscript presented in an intelligible fashion and written in standard English?

Reviewer #1: Yes

Reviewer #2: Yes

6. Review Comments to the Author

Reviewer #1: 1.- It is recommended in the summary to clearly identify the following structure: Identification of the problem, research objective, method used, result and main conclusion

2.-The body of the article should be improved by applying the structure of the IMRYD research methodology

3.-The 8 steps to decoding hybrid threats are important to improve the EU space.

Reviewer #2: I am very happy with the revisions made, which I think have really strengthened the paper and addressed the concerns I had. I thank the author for their genuine and reflective response!

7. PLOS authors have the option to publish the peer review history of their article (what does this mean?). If published, this will include your full peer review and any attached files.

Reviewer #1: **Yes: **Moisés Toapanta, PhD.

Reviewer #2: No

---

## [Author Response · Author response to Decision Letter 1]

12 Mar 2024

# GENERAL COMMENTS TO REVIEWERS

We would like to thank the time you devoted to providing insightful recommendations. Once again, your comments have improved our paper and we hope you agree that this revised version builds a stronger contribution. As always, we have taken the revision very seriously. If you feel the revision has fallen short, we can reformulate accordingly. The following paragraphs deal with the detailed comments you raised.

REVIEWER #1

Reviewer point #1: It is recommended in the summary to clearly identify the following structure: Identification of the problem, research objective, method used, result and main conclusion.

Author response #1: Thank you very much for your feedback. We have revised the summary/abstract as per your suggestion, encompassing the identification of the problem, research objectives, methodology, results, and main conclusions. To adhere to the Journal's guidelines, we maintain a maximum limit of 300 words for the abstract and use track changes so that you can easily identify changes throughout the article.

Reviewer point #2: The body of the article should be improved by applying the structure of the IMRYD research methodology.

Author response #2: At the end of the introduction section, we reference our adherence to the IMRyD or IMRaD structure, which is also reflected throughout the article. Hence, the article adheres to the IMRyD format (Introduction, Methods, Results, and Discussion). We have also included two supplementary sections: one providing a theoretical framework and the other offering a conclusive summary focusing on the originality and contributions of our research. These two supplementary sections adhere to well-established academic structures, bolstering the overall strength of the IMRyD framework. We believe that your recommendation enhanced the comprehensiveness and depth of our article, contributing significantly to its scholarly value. Thank you for your valuable input.

Reviewer point #3: The 8 steps to decoding hybrid threats are important to improve the EU space. 

Author response #3: We think this comment is not a revision request. Overall, we deeply appreciate your comments. If you think any of the aforementioned points require further elaboration, we are more than willing to accommodate your suggestions.

REVIEWER #2

Reviewer point #1: I am very happy with the revisions made, which I think have really strengthened the paper and addressed the concerns I had. I thank the author for their genuine and reflective response! 

Author response #1: Dear Reviewer, we want to express our sincere gratitude for your valuable contributions in enhancing our article. Your suggestions and support have been greatly appreciated.

The author,

March, 2024

---

## [Decision Letter · Decision Letter 2]

26 Apr 2024

EU Space Security – An 8-Step Online Discourse Analysis to Decode Hybrid Threats

PONE-D-23-34933R2

Dear Dr. Reis,

We’re pleased to inform you that your manuscript has been judged scientifically suitable for publication and will be formally accepted for publication once it meets all outstanding technical requirements.

Kind regards,

Michal Ptaszynski, PhD

Academic Editor

PLOS ONE

Additional Editor Comments (optional):

Reviewers' comments:

Reviewer's Responses to Questions

**Comments to the Author**

1. If the authors have adequately addressed your comments raised in a previous round of review and you feel that this manuscript is now acceptable for publication, you may indicate that here to bypass the “Comments to the Author” section, enter your conflict of interest statement in the “Confidential to Editor” section, and submit your "Accept" recommendation.

Reviewer #1: All comments have been addressed

Reviewer #2: All comments have been addressed

2. Is the manuscript technically sound, and do the data support the conclusions?

Reviewer #1: Yes

Reviewer #2: Yes

3. Has the statistical analysis been performed appropriately and rigorously? 

Reviewer #1: Yes

Reviewer #2: N/A

4. Have the authors made all data underlying the findings in their manuscript fully available?

Reviewer #1: Yes

Reviewer #2: Yes

5. Is the manuscript presented in an intelligible fashion and written in standard English?

Reviewer #1: Yes

Reviewer #2: Yes

6. Review Comments to the Author

Reviewer #1: Congratulations to the authors, they have made the requested modifications, and I recommend the acceptance and publication of the article.

Reviewer #2: (No Response)

7. PLOS authors have the option to publish the peer review history of their article (what does this mean?). If published, this will include your full peer review and any attached files.

Reviewer #1: **Yes: **Moisés Toapanta, PhD., is a Doctor in Information Technology from the University of Guadalajara- Mexico, Master in Communications Management and Information Technology, MSc. Cum Laude from the National Polytechnic School (EPN), Computer Engineer and Information Sciences, Public Administration Technician, Parachutist and Former Marine. He carried out his doctoral research stays at the Department of Information and Communications Technologies of the Polytechnic University of Cartagena (UPCT) Spain and the Faculty of Systems Engineering of the National Polytechnic School. Principal Professor at the Universitario Rumiñahui - Sangolqui, Guest Research Professor at the Catholic University of Santiago de Guayaquil (UCSG), Director of TFM (Master of Computer Security and Cybersecurity) at UNIR-Spain, Guest Professor in Postgraduate Studies, Director and Co-Director of Master's and Doctorate (PhD) theses at several Universities in Ecuador, Peru, Mexico and Spain. He is a Consultant, Advisor and Mentor in ICT, Information Security Governance, Strategic Alignment, Cybersecurity, Cyberbullying, Projects, IT Infrastructure Administration, Networks and Communications, Strategic Planning, Scientific Research, Training for teaching researchers and professionals. through Global Technology Management "GTM" International. He is an accredited evaluator and researcher, categorized in the RNI- Senescyt, IEEE Author: ID 37086208212.

Reviewer #2: No

---

## [Editor Report · Acceptance letter]

27 May 2024

PONE-D-23-34933R2 

PLOS ONE

Dear Dr. Reis, 

I'm pleased to inform you that your manuscript has been deemed suitable for publication in PLOS ONE. Congratulations! Your manuscript is now being handed over to our production team.

Kind regards, 

on behalf of

Dr. Michal Ptaszynski 

Academic Editor

PLOS ONE